# The Trend of Authentic Leadership Skills in Nursing Education: The Key Role of Perfectionism and Self-Efficacy

**DOI:** 10.3390/ijerph19041989

**Published:** 2022-02-10

**Authors:** Mariusz Jaworski, Mariusz Panczyk, Anna Leńczuk-Gruba, Agnieszka Nowacka, Joanna Gotlib

**Affiliations:** 1Department of Education and Research in Health Sciences, Faculty of Health Sciences, Medical University of Warsaw, Litewska 14/16, 00-518 Warsaw, Poland; mariusz.jaworski@wum.edu.pl (M.J.); joanna.gotlib@wum.edu.pl (J.G.); 2Department of Development of Nursing, Social and Medical Sciences, Faculty of Health Sciences, Medical University of Warsaw, Erazma Ciołka 27, 01-445 Warsaw, Poland; anna.lenczuk-gruba@wum.edu.pl; 3Department of Obstetrics and Gynecology Didactics, Faculty of Health Sciences, Medical University of Warsaw, Litewska 14/16, 00-575 Warsaw, Poland; agnieszka.nowacka@wum.edu.pl

**Keywords:** leadership, nursing, self-efficacy, perfectionism, nursing education

## Abstract

(1) Background: Shaping leadership skills is a complex process, which may be modified by psychological factors such as self-efficacy and perfectionism. The aim of the study was to determine whether perfectionism can be a mediator between self-efficacy, and authentic leadership skills in nursing students; (2) Methods: The cross-sectional study included 615 Polish nursing students (women = 96.3%) was carried out at Medical University of Warsaw in 2019. The following research tools were used: Authentic Leadership Questionnaire, Almost Perfect Scale-Revised (APS-R), and General Self-Efficacy Scale (GSES); (3) Results: The level of perfectionism is a significant mediator of relations between self-efficacy as measured by the GSES and the level of authentic leadership (Sobel test: t = 6.958; *p* < 0.000). The relation, without a mediating factor, is positive, and the standardized beta coefficient for the feeling of self-efficacy totals beta = 0.470 (*p* < 0.000), while in the presence of a mediator the strength of the correlation is smaller and amounts to beta = 0.366 (*p* < 0.000); (4) Conclusions: Personality factors such as self-efficacy and perfectionism play an important role in shaping AL skills of nursing students. Therefore, academic teachers should pay special attention to strengthening students’ self-efficacy and adaptive perfectionism. In this context, adequate feedback and reflection may be important.

## 1. Introduction

In the nursing literature, leadership skills are analyzed in terms of the effectiveness of communication within the therapeutic team as well as the quality of medical care. Thus, leadership skills that would be most effective in clinical work are sought [1,2,3,4,5,6]. Effective leadership in nursing is crucial for patients’ safety [7]. It should be noted that there are many leadership models in nursing (e.g., transformational, transactional, laissez-faire and authentic), which have been modified over time. In the literature, the transformative style of leadership was the most analyzed. However, this type of leadership showed a poor effectiveness [8]. This is related to the changing healthcare systems in many countries. For this reason, researchers started looking for a new style of leadership and turned their attention to authentic leadership (AL) [9,10].

AL deserves special attention, because it is associated with more influence on staff and increased effectiveness of work (e.g., it increases productivity and job performance) [11,12,13]. There are four dimensions to an authentic leader: self-awareness (e.g., the leaders’ knowledge of their strengths and weaknesses), transparency (e.g., sharing information, feelings, and attitudes), morals and ethics (e.g., it reflects the leaders’ behavior in line with the standards of internal moral conduct), and balanced processing (e.g., it concerns proper decision making after hearing the opinions of their followers) [11,12,13]. All of these AL dimensions may bear great significance in clinical work [5]. AL is connected not only with work organization, but may increase the effectiveness of the communication process between members of the therapeutic team. Clinical work demands team work which entails cooperation with numerous specialists. Introducing AL in the workplace can improve the quality of work (e.g., patient safety), the nurses’ well-being and prevent burnout [3,10,14,15]. It is related to shaping a positive, safe and friendly work environment [14,15].

Leadership skills are not permanent personality traits as they are social skills. Therefore, they must be properly shaped through social interaction [12,13]. In this process, personality traits play an important role; although they are relatively stable, they are still able to develop throughout one’s adulthood. This makes the process dynamic. It is impossible to separate personality from social functioning. These two dimensions are closely related to each other. It is related to the role-based perspective on the adaptive nature of personality during the transition from the one role to other. However, the relationship between personality traits and social skills can have different effects on behavior. It depends on the configuration [16].

Similarly, it can be assumed that a similar relationship exists between personality traits and AL skills. Personality traits can influence the expression of AL skills. Therefore, it seems justified to analyze those personality traits that have the greatest impact on the expression of leadership skills. The challenge of finding the relationship between AL and personality traits is the subject of study [17]. It is stressed that the key is to look at AL from the perspective of predisposition factors, but not results. In other words, it makes sense to focus on those factors that may positively influence AL development. The research results which clearly indicate the positive effects of AL in the work environment are given as the justification for this approach [3,10,14,15]. According to Shahzad et al. [18] the personality traits of a leader can predict his/her authentic leadership style. This is an important observation because the result of study, using the Five-Factor Model of Personality (FFM), emphasizes a direct relationship between personality and AL skills. FFM is one of the most known models for conceptualizing personality traits. FFM characterizes people in terms of five important personality traits. They are: extraversion (also often spelled extroversion), agreeable-ness, openness, conscientiousness, and neuroticism. In the context of FFM, it was noted that extraversion, agreeableness, conscientiousness and openness to experience can have a positive impact on shaping the authentic leadership style, while neuroticism is negatively related to it [18]. This observation provides the basis for looking for a link between personality traits and AL skills on a more detailed level.

It should be noted that self-awareness plays a special role in the concept of AL [11,12,13]. It is a dimension directly related to the perception and attitude toward one’s own skills. Self-awareness and positive perception of one’s own skills is important for the personal and professional development of nurses, and for the development of an effective nurse–patient relationship [19]. Furthermore, high self-awareness is claimed to lead to better decision making, and is linked to team performance and authentic leadership [20]. In this context, personality traits related to self-awareness, which directly influence the effectiveness of human activities, become more important. These include self-efficacy [21] and perfectionism [22].

The literature emphasizes the existence of a relationship between perfectionism and self-efficacy or self-compassion in the context of assessing the effectiveness of actions [22]. The interaction of these two traits can affect self-confidence. A person with a high level of self-efficacy will be willing to undertake various activities [23].

Self-efficacy is the belief that one can perform novel or difficult tasks, or cope with adversity in various domains of human functioning [21,24]. The construct self-efficacy is broadly discussed in the literature as a psychological variable, which refers to the human sense of confidence and competence. It also refers to human judgments of what people think they can do. Thus, this trait determines the behavior of a person that relates to his/her beliefs about themselves [24,25].

In the context of perfectionism, it has been observed that a person with high perfectionism will set himself high expectations [23]. Perfectionism is especially important in nursing, because nursing imposes high standards and perfectionist expectations on professionals during care for patients [22]. Perfectionism is a personality trait that is difficult to define as it has two subdimensions—maladaptive perfectionism and adaptive perfectionism. It should be noted that these two subdimensions of perfectionism are not on one continuum. They need to be treated as two different subdimensions. The maladaptive perfectionism is characteristic for persons who create extremely high and unrealistic goals, which often leads to failure. It does not have a positive effect on the development of that person. Such a person may focus on details of an activity too much, which totally paralyzes its execution. Such a situation is called maladaptive perfectionism. The second dimension, adaptive perfectionism, should help perform activities properly and meticulously [26].

The element that connects self-efficacy and perfection is the aim of activity, which is focused on positive results, although, the form of achieving these results may be different [23]. There are studies that emphasize that the direction of this relationship may depend on the type of perfectionism. A negative relationship can be observed in the context of the relationship between self-esteem and maladaptive perfectionism. However, it is noted that there is a positive relationship between self-esteem and adaptive perfectionism [27]. Another important common element is their influence on social skills, e.g., coping with stress and interpersonal communication [28]. Consequently, self-efficacy and perfection are in a close relationship with social skills. Considering this context, it is important to pay attention to the relationship between these two personality traits in the context of social skills such as AL skills. There is a lack of studies that analyze the complex relationship between perfectionism, self-efficacy and AL skills. There are studies that analyze these psychological variables separately and not as a complex [16,17,18,29].

The close relationship between AL skills and self-efficacy is justified in the literature focused on nurses [29]. It has been observed that self-efficacy has a positive effect on shaping leadership skills. This is related to the fact that self-efficacy has a direct positive relationship with self-awareness, (i.e., the leaders’ knowledge of their strengths and weaknesses) [10]. Furthermore, a positive correlation between AL and self-efficacy was observed in the new graduate nurses group. It has been reported that it is leading to an increase in confidence in their ability to manage work-related challenges [30]. In the context of these data, it is important to verify this relationship in the group of nursing students. There is a dearth of studies analyzing the relationships between nursing students.

The relationship between perfectionism and AL skills is less documented in the literature. However, there is indirect evidence suggesting a positive role of perfectionism in shaping the discussed competences in medical students [31]. In another study, the relationship between perfectionism (self-oriented, socially prescribed, and other-oriented perfectionism) and three types of self-rated leadership behavior (Monitoring, Transformational, and Servant Leadership) was noted. However, this relation is not clear and requires a deeper insight [32]. It should be noted that this has not been verified in relation to nursing students.

There are limited studies analyzing the relationship between perfectionism and self-efficacy for shaping or improving leadership skills in nursing. An association between perfectionism and self-efficacy with regard to leadership skills was found between emergency medicine students [31] and the empirical study conducted with German leaders [32]. For this reason, studies looking to determine the mechanisms of a positive relationship between self-efficacy and leadership skills with high maladaptive perfectionism are required.

### 1.1. Theoretical Assumptions of the Model

The analysis of the relationship between self-efficacy and AL among nursing students can be used in the modification of curricula, and thus is a better preparation of these students for clinical work. It should be noted that there is a positive relationship between self-efficacy and AL [10,33]. However, scientists emphasize that this relationship may be modified by various factors (e.g., psychological features). Taking into account the nature of the variable, such as perfectionism [26,27,28], it may be assumed that it will be a mediator of the relationship between self-efficacy and leadership skills (Figure 1). Moreover, the complexity of personality such as perfectionism, the dimension which is directly connected with setting high standards to oneself, was selected for further analysis. The High Standards taps into having high expectations about one’s performance and achievements [26]. Such an attitude may have particular significance for acquiring and improving one’s own leadership skills. In this model, AL is the dependent variable, while self-efficacy is the independent variable. Self-efficacy is an independent variable because it is a psychological trait that characterizes every human being. It is a relatively constant variable that is difficult but possible to modify [12,34]. The AL skills are much easier to modify under the influence of internal factors (e.g., personality traits of a nursing student) and external factors (e.g., the environment in which a person functions). Accordingly, AL was considered a dependent variable.

With regard to the theoretical model, the following research hypotheses were formulated:

**Hypothesis** **1** **(H1).**
*The self-efficacy (GSES: independent variable) influences on the level of authentic leadership (ALQ: dependent variable).*


**Hypothesis** **2** **(H2).**
*The self-efficacy (GSES: independent variable) influences on the level of perfection standard.*


**Hypothesis** **3** **(H3).**
*The independent variable (GSES) and mediator (perfection standard) significantly influence the dependent variable (ALQ).*


### 1.2. Aim of Study

The aim of the study was to determine whether perfectionism could be treated as a mediator between self-efficacy, and AL skills in nursing students. Determining whether perfectionism, as a mediator, may enhance or weaken the relationship between self-efficacy and leadership skills was important. Preliminary studies, with a group of emergency medicine students, showed that perfectionism can modify the correlation between self-efficacy and leadership skills [31]. This relationship has not been verified in nursing students.

## 2. Materials and Methods

### 2.1. Study Design

A cross-sectional study design included 615 nursing student respondents. The studies were conducted in two students’ groups. The first group included BSc studies (undergraduate), and the second group MSc studies (postgraduate). The study was conducted between October to December 2019. Non-participant details were not kept. The number of students who denied participating in the study was collected.

### 2.2. Sample

The convenience sampling was used as a method of a selection sample. It was a non-randomized sampling method. It was assumed that the sample size would be the entire student population of the nursing program at MUW.

The criteria for inclusion in the study were: (1) the status of a student of nursing at MUW, (2) participation informed consent.

All nursing students could participate in the study, regardless of the year of study. Undergraduate students (*n* = 353) and postgraduate students (*n* = 262) participated in the study. Among undergraduate students, 169 were 1st year students, 134 2nd year students and 50 were 3rd year students. In the case of postgraduate students, 129 were 1st year students, and 133 were 2nd year students. In the academic year 2017/2018, 911 students studied nursing; of these there were 549 undergraduate students and 362 postgraduate students. The ratio of filled-in questionnaires from undergraduate students was 64.30% and 72.38% for postgraduate students.

### 2.3. Data Collection Procedure

Before commencing the study, consent for conducting the cross-sectional study was obtained among nursing students at the Medical University of Warsaw (MUW). Next, a person responsible for didactics in nursing (coordinator of nursing) was contacted to enable the execution questionnaire survey among students. In this study, the auditorium questionnaire was used. This type of questionnaire is completed by a group of people gathered in one place. However, each participant completes the questionnaire independently. The coordinator, who was part of the research team, was in charge of the accuracy of conducting the research and received instructions for data collection supervision.

The data were gathered from October to December 2019 at the MUW. The principal investigator did not conduct classes with any groups of students. He came at the end of the class and presented the purpose of the study and the possibility of participating in the study in the presence of the teacher conducting the classes. Students who expressed their willingness to participate in the study stayed in the classroom after the classes. Students filled in the survey in the presence of the coordinator.

Research tools (questionnaires) were placed in envelopes. Next, the coordinator distributed them. The students put the completed or incomplete questionnaires in an envelope and left it on their desks. The coordinator then collected the envelopes with the questionnaires after the time designated to complete questionnaires. The mean time of data collection was 10 min. Therefore, the coordinator could not verify the complementarity of the questionnaires and check the number of participants who took part in the study. People who did not want to participate in the study also returned the envelopes, but there were blank questionnaires inside. Such action allowed the students to remain fully anonymous and voluntarily participate in the study. There were no third parties present during data collection. The students were informed of the aim of the study, as well as on the possibility of not participating. The coordinator assured the students full anonymity in the study.

Before data collection, the coordinator attended training conducted by a psychologist. This training had two purposes. First, the research tool concerned the realm of psychological functioning. The potential influence of outsiders needed to be minimized for the respondents answering questions. Second, the coordinator was trained to identify students who felt insecure or distressed about participating in the study, and to provide adequate support. The students could also benefit from psychological support as a psychologist was a member of the research team.

### 2.4. Research Tools

The following research tools were used: Authentic Leadership Questionnaire (ALQ) [12], Almost Perfect Scale-Revised (APS-R) [26], and General Self-Efficacy Scale (GSES) [24].

The ALQ is designed to measure the components that comprise AL, and has four scales: Self Awareness, Transparency, Ethical/Moral, and Balanced Processing. The Polish version of the ALQ was used and consisted of 16 items. The respondent replied to each question using a five- point scale (not at all, once in a while, sometimes, fairly often, frequently, if not always) [12]. The psychometric properties of the ALQ have been validated through a study by Sierpińska and the Cronbach’s alpha is 0.80 [35]. In this study, the Cronbach’s alpha is 0.833 for the ALQ.

The APS-R included 23 items, and contains three variables: (1) High Standards—it is the tendency to set high expectations for oneself and to push oneself to work hard to attain those expectations; (2) Order—it is the tendency to prefer organization, and order in one’s environment and physical surroundings; and (3) Discrepancy—it is the tendency to feel that one is not meeting one’s standards and expectations [26]. Respondents replied to particular questions using a 7-point scale (strongly disagree, disagree, slightly disagree, neutral, agree slightly, agree, strongly agree). The designers of APS-R [26] considered standards, discrepancy and order as defining elements of perfectionism. Both adaptive and maladaptive perfectionists rate highly in Standards and Order, but maladaptive perfectionists also rate highly in Discrepancy. Maladaptive basically means less flexible to the point of frustration and an inability to reach goals or just the continual ‘not good enough’ [26]. The psychometric properties of the APS-R have been validated through a series of studies by Slaney and colleagues and they reported internal consistency estimates from 0.85 to 0.92 [26]. In this study, the Cronbach’s alpha is 0.887 for the APS-R.

The GSES is a self-report measure of self-efficacy and has 10 items. The respondent replies to each question using a 4-point scale (not at all true, hardly true, moderately true, exactly true). The total score is calculated by finding the sum of all the items. For the GSE, the total score ranges between 10 and 40, with a higher score indicating more self-efficacy. The psychometric properties of the GSES have been validated through a study by Juczynski and the Cronbach’s alpha was between 0.76 and 0.90 [24]. In this study, the Cronbach’s alpha is 0.857 for the GSES.

### 2.5. Consent of the Bioethics Committee

The authors of the present study obtained the approval of the Institutional Ethics Committee of Medical University of Warsaw on no contradictions for conducting studies with the use of non-invasive research methods. There was a psychologist on the research team. This was related to the study utilizing psychological factors. Therefore, students who felt uncomfortable because of participating in the study were able to discuss their feelings with the psychologist. Students were informed that the research team was open to their feedback.

### 2.6. Statistical Analysis

All of the statistical analyses were performed using STATISTICA 13.3 (TIBCO©, Inc., Palo Alto, CA, USA). *p*-values < 0.05 were considered to be statistically significant.

Assessment of the prevalence of partial mediation was conducted according to the procedure proposed by Cohen et al. [36] Mediation analysis is made up of three stages, which consists of a series of regression analyses. The parameters of regression function were estimated with the use of the least squares method. Non-standardized (b) and standardized (β) regression coefficients together with 95% confidence intervals were determined.

In the first stage, the influence of self-efficacy (GSES: independent variable) on the level of leadership was evaluated (ALQ: dependent variable). In the second stage the influence of the feeling of self-efficacy (GSES: independent variable) on the level of perfection standard (potential mediator) was assessed. In the third and last stage of the analysis, it was evaluated whether the independent variable (GSES) and mediator (perfection standard) significantly influenced the dependent variable (ALQ). It was hypothesized that the mediation was significant when intermediate relations of the independent variable and the mediator, as well as mediator and dependent variable, were statistically significant. In such cases the determining factor was the result of the Sobel test, which evaluates whether the product of non-standardized regression coefficients for both relations was significantly different to zero [37].

## 3. Results

### 3.1. General Characteristics of the Respondents

The mean age of the respondents was 24.26 years (SD = 4.721). The youngest student was 20 and the oldest was 53. Women prevailed in the analyzed group (96.3%; *n* = 592). Men constituted 3.7% of all respondents (*n* = 23). Students from very large cities dominated in the study (*n* = 243; 39.5%) and villages (*n* = 183; 29.8%). Furthermore, there were students from small towns (*n* = 98; 15.9%) and medium-sized towns (*n* = 66; 10.7%). The smallest share of students came from large cities (*n* = 25; 4.1%).

### 3.2. The Profile of Respondents in Terms of AL Skills and Analyzed Psychological Variables

Table 1 presents the profile of nursing students in terms of AL skills and subdimensions of the skills, with the classification into type of studies (undergraduate vs. postgraduate studies). Undergraduate and postgraduate students did not differ when it came to the intensity of leadership skills (t = −0.900; *p* > 0.05) and all analyzed dimensions of perfectionism: perfection standards (t = −1.500; *p* > 0.05), perfection other (t = −1.635; *p* > 0.05) and perfection discrepancy (t = 1.963; *p* > 0.05). Differences were noted only in relation to self-efficacy (t = −2.662; *p* = 0.008).

Overly high results at the STANDARDS scale showed that the perfectionism has the values from 25 to 49 points. In the analyzed group, 95.6% (*n* = 588) nursing students had a high score on the scale. There were no data recorded in this scope for 1.8% (*n* = 11). In the case of subscale ORDER, results showing perfectionism are values from 4 to 28 points. In the analyzed group there were 92.8% (*n* = 571) of nursing students with high scores on the scale (1.0%, *n* = 6 were not recorded). In the case of subscale DISCREPANCY, results showing perfectionism had values from 42 to 84 points. In the group analyzed there were 60.8% (*n* = 374) nursing students with a high score on the scale (2.4%, *n* = 15 were not recorded). Maladaptive perfectionism was identified in 361 students (58.7%) who featured high scores on the subscales STANDARDS and DISCREPANCY. In other words, over half of nursing students featured maladaptive perfectionism. The maladaptive perfectionism may have a negative impact on the functioning of students, their activities and the results related to the educational process [23,27,28].

### 3.3. Results of Mediation Ansalysis

As no differences were noted between undergraduate students and postgraduate students in relation to general aggregation of leadership skills and perfectionism, further analyses were conducted on the whole group without any further classification into study type. The only difference was noted in relation to self-efficacy; however, the effect size was small (Cohen’s *d* = 0.22). For this reason, an analysis of the nursing students was made together, irrespective of study type, in terms of self-efficacy.

The first of the tested linear regression models describing the influence of the sense of self-efficacy on AL skills was statistically significant (F = 163.714, *p* < 0.001, SS model = 7043.117, MS model = 7043.117, R^2^_adjusted_ = 0.22) and well fitted to the data (Ramsey RESET test, F = 0.107, *p* = 0.899). The results of regression analysis demonstrated that the sense of self-efficacy (β = 0.470, *p* < 0.001) was the factor positively influencing AL skills for the examined nursing students (Table 2). The higher the sense of self-efficacy, the higher the level of AL. The sense of self-efficacy clarifies 22% variability of AL skills.

The second of the tested linear regression models describing the influence of the sense of own efficacy on perfectionism (dimension connected with high standards) was also statistically significant (F = 91.637, *p* < 0.001, SS model = 3700.772, MS model = 3700.772, R^2^_adjusted_ = 0.13) and fitted well with the data (Ramsey RESET test, F = 1.272, *p* = 0.281). Results of the regression analysis demonstrated that the self-efficacy (β = 0.366, *p* < 0.001) positively influenced AL skills (Table 3). The higher the sense of self-efficacy, the higher the level of perfectionism in the dimension of high standards. The sense of self- efficacy explains 13% variability of the analyzed form of perfectionism.

The mediation analysis has shown that both the sense of self-efficacy and perfectionism (the dimension of high standards) independently influence the AL skills of the students. The model was statistically significant (F = 109.960, *p* < 0.001, SS model = 8755.282, MS model = 4377.641, R^2^_adjusted_ = 0.28) and fitted well with the data (Ramsey RESET test, F = 0.288, *p* = 0.750). It was observed that students with a higher level of self-efficacy also had a higher level of AL (β = 0.385, *p* < 0.001). Similarly, students who had higher level of perfectionism (high standards dimension) also had higher level of AL skills (β = 0.251, *p* < 0.001) (Table 4). The higher the level of self-efficacy and perfectionism, the higher the level of leadership. These two variables explain 28% variability of AL skills. The mediation analysis conducted demonstrated the absence of full mediation.

Considering demonstration of the absence of full mediation, testing of significance of partial mediation was performed (Sobel test). The level of perfectionism was a significant mediator of relations between the self-efficacy (GSES) and the level of AL (Sobel test: t = 6.958; *p* < 0.001). The relation, without an intermediary factor, is positive, and the standardized regression coefficient for self-efficacy was β = 0.470 (*p* < 0.001), while in the presence of a mediator the strength of the relation was smaller β = 0.366 (*p* < 0.001).

## 4. Discussion

The first regression result showed a positive relationship between self-efficacy and AL skills. This relationship is in line with the general concept of AL [11,12,13]. According to this concept [11,12,13], self-awareness of skills as well as strengths and weaknesses is a key dimension of AL. Self-awareness is an important factor in self-development and skills improvement. Moreover, this may affect the quality of patient care quality. It is emphasized that the dimensions of authentic leadership, such as self-awareness, predict patient care quality [38]. The results of studies in the group of nurses confirmed a relationship between self-efficacy and AL skills [10,19]. Our study also confirms this relationship in the group of nursing students.

It should be noted that high self-efficacy allows people to develop AL skills more effectively. In turn, a high level of AL skills allows people to create a positive work environment. This is confirmed by research [17,18,39,40,41]. Similar observations were also made in relation to nurses [14,15,42,43,44], as well as for nurses starting work [45]. It confesses the strong relationship between self-efficacy, and leadership skills [10,31]. This study also confirms the relationship.

The second regression result showed a positive relationship between self-efficacy and perfectionism. This is an interesting observation in the context of the approach to perfectionism. There are numerous approaches to perfectionism. Perfectionism is a multidimensional concept which includes striving for flawlessness and setting high goals [26]. The discussed variable may be viewed in three important dimensions such as: self-oriented perfectionism, other-oriented perfectionism, and socially prescribed perfectionism [46]. The perfectionism can be an important personality trait in the health care system. Additionally, society expects its nurses to be flawless and to do their job to the best of their ability [47]. This results in that recognizing perfectionism on a personal level is important for nurses’ health and work performance. Patients expect full professionalism from medical personnel, and maladaptive perfectionism may hamper this [48].

In the context of health care, self-oriented perfectionism may be of high significance. This type of perfectionism is defined as an intrapersonal dimension that involves requiring perfection of oneself, constantly striving to achieve unrealistically high standards, and critically evaluating one’s own performance [46].

If we look at perfectionism in greater detail, adaptive perfectionism is great importance here [39,40,48]. The literature highlights that nurses and their workplace colleagues are all susceptible to maladaptive perfectionism [48]. Kelly [49] showed that nursing students show a higher incidence of maladaptive perfectionism than the general population. Maladaptive perfectionism interferes with every aspect of an individual’s life, and a nurse’s career [49]. In this regard it is reasonable to undertake work which would further help to understand this mechanism. This especially relates to depression-prone perfectionism that is reflected in behavior such as consistently setting unrealistic goals, ruminating, never feeling satisfied with one’s own or others’ performance, avoiding being judged by others, and declining any help required [48].

It should be noted that the study presented also demonstrates that the problem of maladaptive perfectionism concerns a large number of nursing students. This confirms the need to modify and adapt the curricula so that it includes programs for supporting students with high maladaptive perfectionism. The literature highlights the need for academic teachers to become educated on perfectionism, and to help students at risk by providing resources [49]. However, it should be noted that the number of studies addressing perfectionism among nursing students is limited [48,49], and accounts for a substantial research gap.

The third regression result showed that perfectionism may perform an important role as a mediator between self-efficacy, and AL skills in nursing students. The reported observation confirms that the relationship between self-efficacy, perfectionism and AL skills is not only typical for emergency medical students [31], but also for nursing students. It also emphasizes that the theoretical assumptions were valid and were confirmed. Furthermore, there are no studies in the literature that analyze the relationship between perfectionism, self-efficacy and AL skills. Therefore, the presented study cannot be compared to the results of other researchers.

It can be assumed that a student who obtains a diploma in nursing should feature not only substantive knowledge, but also psychosocial skills important for clinical work; for example, AL skills. Furthermore, it is not enough to be confident and convinced about the effectiveness of one’s own actions in relation to the objective set. It is important to see how this self-confidence is modified by personality traits. In this context and results of present model, perfectionism becomes a link between self-efficacy and social skills, such as AL skills.

Setting high objectives and the connected standards of conduct is directly related to experiencing various emotional conditions. Some people set high standards for themselves, but can also experience positive emotions. These individuals engage in the “relaxed and careful” pursuit of activities and evaluate themselves against high but reasonable self-standards. This is the so-called healthy and positive form of perfectionism [46]. It is believed that such a form of perfectionism should be enhanced, as not only does it enable consistent striving to achieve the goal and performing tasks reliably, but it also helps effectively cope with emotions or stress [49]. Such a skill may also be particularly important in clinical work. In contrast, some persons with a high level of perfectionism may experience mainly negative emotions. These people are profiled as a maladaptive, unhealthy, negative form labelled “neurotic”. These individuals engage in “tense and deliberate” pursuit of unreasonable expectations [46]. The key difference is that the adaptive perfectionists derive pleasure from their striving, but maladaptive perfectionists “never seem to do things good enough to warrant that feeling” [46]. To sum up, it can be assumed that positive emotions related to perfectionism can enhance the effect of self-efficacy and can be helpful in shaping AL skills. Therefore, this relationship should be strengthened in the education process.

The results obtained have important implications in the modern education of nurses in the context of shaping leadership skills. The developed model can be used in designing practical classes focused on developing and strengthening leadership skills. It is known that shaping leadership skills can increase the efficacy of clinical work (e.g., patient safety or reducing the number of medical errors). Furthermore, the appropriate shaping of these skills influences the functioning in the work environment, especially job satisfaction [43,44]. The leadership skills are classified as social skills. Therefore, they can be shaped through interpersonal relationships. Mentoring can be used in this case, because this method allows students to develop their social skills (e.g., leadership skills) based on personality dispositions. As shown in the developed model, personality traits are important in this process. Initial diagnosis of such personality traits as perfectionism [39] and self-efficacy [19] may be especially useful, as they play a key role in acquiring these skills. Such a diagnosis would allow for the choice of an adequate and individual teaching strategy, but it also requires a greater elasticity of the teacher (trainer–mentor) in the selection of teaching training methods. In this context, an approach based on mentoring may be critical.

In this teaching process, reflection can play an important role. It allows the student to assess their strengths and weaknesses, and take appropriate action to improve their own skills [17,18]. Further research examining the role of reflection in improving leadership skills should be conducted. It is important to take into account the relationship between self-efficacy and perfectionism in this context.

### 4.1. Limitations of the Study

The presented study has certain limitations. It should be interpreted with caution. In the first place, this is a cross-sectional study design. This excludes any conclusions concerning the direction of changes in time. It seems justified to undertake longitudinal studies that could overcome this problem. The presented study should be treated as an initial exploration to advise future longitudinal studies, which can consider the role of perfectionism as the mediator between self-efficacy and AL skills not only in students, but also among professionally active nurses. Secondly, in the study a convenient sample, data from only one university was used. This may limit generalization of the results to a wider population of students. Another limitation is the impossibility of a comparison of the obtained results with other studies. In the presented study, no additional mediators or moderators, such as negative thinking or emotional control, were considered. Future research should investigate the influence of the variables as potential mediators or moderators. It should be noted that the study was realized at the beginning of the class. For this reason, the power difference between student nurses and their lectures can influence the students’ feelings related to voluntary participation. However, students were assured that participation in the study or refusal to participate in the study did not affect the teaching process. The students could stop the study at any time without consequence.

### 4.2. Strengths of the Study

Determining the role of perfectionism as the mediator in the relation between self-efficacy and AL skills is a strength of the study. In the literature, there is an absence of studies which would undertake the topic not only among nursing students but also professionally active nurses. Additionally, the obtained research results indicate that psychological variables, such as personality traits, need to be examined altogether. Personality traits do not function separately, but instead interact. It is especially relevant in relation to shaping psychosocial skills such as AL skills. Another important strength of the study is the group of respondents which comprised nursing students. The group of analyzed students accounted for 67.51% of all nursing students studying at the MUW.

## 5. Conclusions

Personality factors (self-efficacy and perfectionism) play an important role in shaping the AL skills of nursing students. Regression analysis showed that there was a correlation between self-efficacy and perfectionism. The higher the sense of self-efficacy, the higher the level of perfectionism in the dimension of high standards. These results showed that the interaction of self-efficacy and perfectionism can have an impact on the professional development of nursing students, e.g., developing social competences. Therefore, academic teachers and teachers of practical skills during clinical practice should pay special attention to strengthening students’ self-efficacy and adaptive perfectionism. Strengthening these two personality traits will help you develop AL skills more effectively. These are salient observations in the context of teaching these skills and can be used in their future clinical practice. In this context, adequate feedback and reflection may be important. The introduction of courses to develop AL, self-efficacy and perfectionism for nursing students into the curriculum should be considered.

## Figures and Tables

**Figure 1 ijerph-19-01989-f001:**
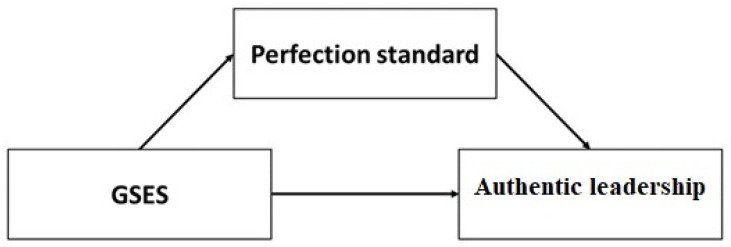
Theoretical model.

**Table 1 ijerph-19-01989-t001:** The profile of respondents in terms of authentic leadership skills and perfectionism.

Variable	N	Min–Max	Mean	Standard Deviation
**Authentic Leadership Skills**				
Balanced_Processing	605	5–15	11.85	2.065
Bachelor’s degree studies	348	5–15	11.819	2.080
Master’s degree studies	256	6–15	11.906	2.042
Self_Awareness	610	8–20	14.89	2.377
Bachelor’s degree studies	349	9–20	14.72	2.382
Master’s degree studies	260	8–20	15.11	2.358
Transparency	609	11–25	18.83	2.858
Bachelor’s degree studies	350	11–25	18.82	2.837
Master’s degree studies	258	12–25	18.83	2.892
Moral	607	9–20	16.28	2.318
Bachelor’s degree studies	351	9–20	16.23	2.297
Master’s degree studies	255	9–20	26.36	2.352
Leadership (Global score)	588	40–80	61.86	7.417
Bachelor’s degree studies	341	40–78	61.64	7.340
Master’s degree studies	247	41–80	62.16	7.520
**Perfectionism**				
Perfection standards	604	7–78	38.78	6.787
Bachelor’s degree studies	342	11–78	38.42	6.680
Master’s degree studies	262	7–49	39.26	6.900
Perfection order	609	4–28	21.64	4.343
Bachelor’s degree studies	346	6–28	21.39	4.000
Master’s degree studies	263	4–28	21.97	4.750
Perfection discrepancy	600	12–84	47.57	15.103
Bachelor’s degree studies	341	12–84	48.62	14.760
Master’s degree studies	259	12–84	46.18	15.460
**The sense of Self-Efficacy**				
Self-efficacy (Global score)	603	15–40	29.99	4.212
Bachelor’s degree studies	345	16–40	29.60	3.993
Master’s degree studies	257	15–40	30.49	4.435

**Table 2 ijerph-19-01989-t002:** The influence of the sense of one’s own efficacy on authentic leadership skills.

Independent Variables	b	β_std_	Confidence Interval		t	*p*-Value
			−95%	+95%		
Intercept	36.766	-	-	-	18.574	<0.001
The sense of self-efficacy	0.838	0.470	0.398	0.543	12.795	<0.001

b—unstandardized regression coefficient, β_std_—standardized regression coefficient.

**Table 3 ijerph-19-01989-t003:** The influence of the sense of one’s own efficacy on perfectionism.

Independent Variables	b	β_std_	Confidence Interval		t	*p*-Value
			−95%	+95%		
Intercept	21.032	-	-	-	11.224	<0.001
The sense of self-efficacy	0.593	0.366	0.291	0.442	9.573	<0.001

b—unstandardized regression coefficient, β_std_—standardized regression coefficient.

**Table 4 ijerph-19-01989-t004:** Mediation analysis.

Independent Variables	b	β_std_	Confidence Interval		t	*p*-Value
			−95%	+95%		
Intercept	30.584	-	-	-	30.584	<0.001
Perfectionsm (High Standards)	0.275	0.251	0.177	0.326	6.607	<0.001
The sense of self-efficacy	0.686	0.385	0.311	0.460	10.130	<0.001

b—unstandardized regression coefficient, β_std_—standardized regression coefficient.

## Data Availability

Not applicable.

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
