# Peer review of "The Trend of Authentic Leadership Skills in Nursing Education: The Key Role of Perfectionism and Self-Efficacy"

_ijerph, 2022, doi:10.3390/ijerph19041989_

Round 1
Reviewer 1 Report
Dear Authors,
Thank you for the opportunity to read your work. These are interesting results that may have implications for better preparation for the nursing profession. Increasing self-esteem and the effectiveness of interventions are important for both nursing organization and patient satisfaction with care.
Despite a good evaluation of the manuscript I have a few comments:
1 - 2.2.1. Theoretical assumptions of the model - it seems to me that posting this section here is not relevant. Maybe it would be better to move it to the Introduction?
2 - The study exclusion criteria can be removed as they are analogous to the inclusion criteria
3 - 2.2.2 Data collection procedure needs to be rewritten. It is very good that the principal investigator was trained by a psychologist etc. However, this section should include a description of the research process, when were the data collected, during class time? Did the principal investigator conduct classes with any groups of students? Were the questionnaires in envelopes?
4 - 2.2.3 Setting - earlier the authors did not write about the stages of the study. The authors probably meant two levels of education? Information about nurse education is not needed. The mentioned EU directives and ministerial regulations are not related to the manuscript topic and part of the paper so they should be removed.
5 - Institutional Review Board Statement, reference [45] and lines 453-456 can be removed
6 - References:
30 and 31 have incorrect notations
45 refers to Institutional Ethics Committee approval and is not needed in the list
Reviewer 2 Report
Dear authors, it is a great pleasure to read this very well written paper. The aim was clear and the method/material too. Please, declare the study design because it is not noted neither in the abstract, nor in the full text. Results, Discussions are fully explained. I think that it is a ready to publish paper, adding new elements in the literature.
Reviewer 3 Report
- Very interesting topic related to perfectionism and self-efficacy in authentic leadership amongst nursing students
- Would be interesting to replicate the study amongst experienced nurse leaders
Introduction
- Correction (lines 78-80) "It also refers to human judgments of what people think that they (their) can do. Thus, this trait determines the behavior of a person that relates to his/her beliefs about themselves (himself).
2.1 Materials
- Rephrase for clarity (line 118) "Students of all years at the discipline nursing"
2.2.1 Theoretical assumptions of the model
- It would be helpful if all perfectionism dimensions would be defined (high standards, order and discrepancy)
- Redundancy (line 142-143) "The High Standards taps into having high standards..."
- Correction (line 149) "(...) under the influence of internal (external) factors..."
2.2.2. Data collection procedure
- Rephrase for clarity (line 158) "(...) the execution an auditorium questionnaire."
2.2.4 Research tools
- If standards and discrepancy are defining elements of perfectionism, what is the role of Order as a third variable?
3. Results
- Clarify (line 268) what is the subscale OTHERS?
4. Discussion
- Rephrase for clarity (line 341-342) "(...) avoiding evaluation by important others..."
- Clarify (line 357) Does "high negative perfectionism" relate to maladaptive perfectionism?
- The implications are difficult to understand as they are currently presented (lines 385-392) - requires reformulation
4.2 Strengths of the study
- Line 430 ends with a "5." that should be deleted.
References
- In section 2.2.4 (research tools), skips from reference 33 to reference 35.
- Only 12 of 45 references date 5 years or less. Can you integrate more recent sources?
Reviewer 4 Report
Dear authors,
It was a pleasure for me to read your article. Especially the method you use and the results you find are scientifically valuable and contribute to the field. For this reason, I congratulate you. However, I have to say that the same success was not achieved especially in the introduction and discussion sections. Redesigning these two sections will make your article more qualified. My recommendations are presented below.
I wish you good work.
Kind regards.
The following points should be noted in the article:
In the abstract section, information about when and where this research was conducted and the number of participants should be given.
Although the introduction is short, it is complex and weak. The introduction is designed in separate sections. Although the three subjects that the research deals with are included in the introduction, it is not explained what kind of relationship there is/may be between them. Since it is mentioned that there are few studies on this subject in the literature, these studies should be referenced and the relationships between authentic leadership, self-efficacy and perfectionism should be emphasized. The results of studies examining the relationship between these topics, even in areas other than nursing, should be shown.
In the introduction, more emphasis is placed on authentic leadership, but the topics of self-efficacy and perfectionism are not adequately explained. A more in-depth treatment of these two concepts is important for ensuring integrity.
It may also be helpful to look at the following resources:
- Duygu Hiçdurmaz, Adeviye Aydın. The Relationship Between Nursing Students' Self-Compassion and Multidimensional Perfectionism Levels and the Factors That Influence Them. J Psy Nurs. 2017; 8(2): 86-94. 10.14744/phd.2017.40469
- Ling-Na Kong, Li Yang, Yi-Nan Pan, Shuo-Zhen Chen. Proactive personality, professional self-efficacy and academic burnout in undergraduate nursing students in China. Journal of Professional Nursing. 2021; 37: 690-695.
1.1. Aim of study. Although the purpose of the study and the gap it will fill in the literature are written in various sections in the introduction, they are not presented here. In the introduction, the absence of research in this context in the field of nursing is mentioned, but more emphasis should be placed on why it is necessary. Here, the importance of research should be explained in a way that satisfies the reader.
2.1. Materials. Although it is stated that the response rates of 64% and 72% are sufficient, it is not specified whether the sample was selected by simple random method or stratified method. Both the method and the reason should be clearly stated.
2.2.1. Theoretical assumptions of the model. Research design is well suited to hypotheses. The nature of the study and the results obtained require hypothesis formulation. Giving hypotheses in the introduction or method section, and explaining the confirmation of the hypotheses in the findings and discussion sections will enrich the article design.
2.2.4. Research tools. The Cronbach's Alpha coefficients in the validation studies of the scales are given, but their values ​​in this study are not stated. The Cronbach Alpha value of all three scales in this study should be stated.
Page 7, lines 274-276. Were that preventing things from going smoothly in this research, or was it preventing things from going smoothly at the hospital in general? Further findings with in-depth analysis are needed to make this comment!
The discussion section should be redesigned in order to deal with the regression results in order and to compare them with similar research results in the literature. Because in the first paragraph of the discussion, the result of perfectionism was given and compared with the literature. This is true. However, after giving the result of self-efficacy in the second paragraph, a comparison was made with the results of perfectionism in the literature. The discussion section should be handled more systematically. Much has been commented on perfectionism, but not as much on authentic leadership and self-efficacy.
Page 10, lines 385-388. What kind of a relationship is established between the results of this research and increasing the number of nurses or job satisfaction? The basis for this link was weak and is irrelevant to this research.
In the conclusion section, it would be better to briefly include other results, not just the mediating role of perfectionism and self-efficacy in shaping AL skills. A brief explanation of all three regression analysis results in this section and appropriate recommendations make the conclusion section ideal.
Round 2
Reviewer 4 Report
Dear authors,
I think your corrections are on point and good. You did a good job. Congratulations.
Kind regards.